# Post-Proline Cleaving Enzymes (PPCEs): Classification, Structure, Molecular Properties, and Applications

**DOI:** 10.3390/plants11101330

**Published:** 2022-05-18

**Authors:** Anis Baharin, Tiew-Yik Ting, Hoe-Han Goh

**Affiliations:** Institute of Systems Biology, Universiti Kebangsaan Malaysia, UKM, Bangi 43600, Malaysia; anis.baharin@gmail.com (A.B.); tingtiewyik@gmail.com (T.-Y.T.)

**Keywords:** prolyl endoprotease, prolyl oligopeptidase, protease, protein engineering, proteolytic enzyme, proteomics, therapeutics

## Abstract

Proteases or peptidases are hydrolases that catalyze the breakdown of polypeptide chains into smaller peptide subunits. Proteases exist in all life forms, including archaea, bacteria, protozoa, insects, animals, and plants due to their vital functions in cellular processing and regulation. There are several classes of proteases in the MEROPS database based on their catalytic mechanisms. This review focuses on post-proline cleaving enzymes (PPCEs) from different peptidase families, as well as prolyl endoprotease/oligopeptidase (PEP/POP) from the serine peptidase family. To date, most PPCEs studied are of microbial and animal origins. Recently, there have been reports of plant PPCEs. The most common PEP/POP are members of the S9 family that comprise two conserved domains. The substrate-limiting β-propeller domain prevents unwanted digestion, while the α/β hydrolase catalyzes the reaction at the carboxyl-terminal of proline residues. PPCEs display preferences towards the Pro-X bonds for hydrolysis. This level of selectivity is substantial and has benefited the brewing industry, therapeutics for celiac disease by targeting proline-rich substrates, drug targets for human diseases, and proteomics analysis. Protein engineering via mutagenesis has been performed to improve heat resistance, pepsin-resistant capability, specificity, and protein turnover of PPCEs for pharmacological applications. This review aims to synthesize recent structure–function studies of PPCEs from different families of peptidases to provide insights into the molecular mechanism of prolyl cleaving activity. Despite the non-exhaustive list of PPCEs, this is the first comprehensive review to cover the biochemical properties, biological functions, and biotechnological applications of PPCEs from the diverse taxa.

## 1. Introduction

Peptidases, proteases, proteinases, or proteolytic enzymes break down peptide molecules, mostly by attacking peptide bonds via hydrolysis. Extensive protease studies found seven main catalytic types, which can be categorized by the nucleophiles for the proteolytic reaction with water molecules. The nucleophiles mainly comprise different amino acid residues, including aspartate, cysteine, glutamate, serine, and threonine, whilst another type contains metal ions (mostly zinc) at their active site, denoted as metalloproteases [1]. The seventh catalytic type performs intramolecular self-cleavage without water molecules using asparagine as a nucleophile [2]. Besides catalytic type, proteases can be further classified according to the peptide cleavage sites between the amino (N-) and carboxyl (C-) termini of proteins, such as aminopeptidase, carboxypeptidase, endopeptidase, or exopeptidase [3]. An oligopeptidase only cleaves peptides but not proteins. As of March 2022, there were 281 protein families in the MEROPS peptidase database, comprising 4431 peptidases with identifiers, which were classified into nine catalytic types aspartic (A, 324), cysteine (C, 1059), glutamic (G, 22), metallo (M, 1098), asparagine (N, 24), mixed (P, 54), serine (S, 1728), threonine (T, 105), and unknown/unclassifiable (U, 17).

This review focuses on post-proline cleaving enzymes (PPCEs) from different peptidase families, besides the well-known prolyl endoprotease/oligopeptidase (PEP/POP) from the serine peptidase family. The term PPCEs in this review is used to encompass other proteins with post-proline cleaving (PPC) activity. Based on the 108,005 cleavage site records of the MEROPS database, we used the “What peptidase can cleave this bond?” search function to compile a list of peptidase family showing specific post-proline cleavages and supported by literature searches. Many PPCEs can be found in the MEROPS peptidase database from 17 families with different substrates and proline-cleavage specificities (Table 1). This non-exhaustive list of PPCEs is mostly dominated by the serine protease (S) family, including S06, S09, S15, S28, S33, and S37 with a nucleophilic serine residue for catalysis [4]. Both PEP and POP hydrolyze polypeptides at the C-terminal of proline residues (Pro-Xaa), with PEP not restricted by protein size, while POP can only cleave small peptides [5]. Studies have shown that different PEPs can cleave a 33-mer gliadin peptide from wheat and synthetic peptides at different rates for different species [6,7]. Apart from the serine protease family S9, the metalloprotease families M12 and M13 are also overrepresented with subfamilies possessing the PPC activity (Table 1). To date, reports of PPCEs from plants are relatively scarce compared to more extensive studies of PPCEs in microbes and animals (Table 2). 

Post-proline proteolysis is prevalent given its importance in nature. PPCEs are involved in various biological functions in diverse taxa of organisms including microbes, fungi, animals, and plants. The biological functions described in the section below include microbial invasion through the protective barrier, pathophysiology of human diseases, and plant nutrient uptake/mobilization, growth, and development. Apart from proteolysis, PPCEs can affect the biological system through protein–protein interactions (PPIs), such as the acceleration of α-synuclein aggregation that can lead to Parkinson’s disease [8]. Hence, studies of PPCEs are important for the pharmaceutical industries.

The purpose of this review is to discuss the recent findings on the structure-function analysis of PPCEs to provide insights on the molecular mechanism of PPC activity for further investigations. The biochemical properties, such as substrate specificity and optimal conditions of enzymatic activity were summarized for both the native and recombinant PPCEs from the recent experimental findings. Lastly, the utilization of PPCEs for different biotechnological applications in the food/beer brewing industry, pharmaceuticals/therapeutics, and proteomics will be described with efforts in protein engineering. 

**Table 1 plants-11-01330-t001:** A non-exhaustive list of selected enzymes with post-proline cleaving activity based on records from the MEROPS peptidase database release 12.4.

Family	Subfamily	Peptidase	UniProt/MEROPS ID	Specificity ^†^	Substrate	Cleavage Site	Reference
G3 [1/1]	G03.001	Strawberry mottle virus glutamic peptidase	MER1365461	2/2 (100%)	Polyprotein	Peptide-Pro↓Ala/Lys-Peptide	[9]
M2 [4/7]	M02.003	Peptidyl-dipeptidase angiotensin-converting enzyme (ANCE)	Q10714/MER0001987	6/7 (86%)	Bradykinin	Peptide-Pro↓Phe/Tyr-Arg	[10]
	M02.006	Angiotensin-converting enzyme-2	Q9BYF1/MER0011061	10/14 (71%)	Angiotensin-2	Peptide-Pro↓Phe	[11]
M3 [8/11]	M03.002	Neurolysin	P42676/MER0001942	68/128 (53%)	Neurotensin	Peptide-Pro↓Tyr-Peptide	[12]
M9D [1/2]	M9D.002	Proline-specific peptidyl-dipeptidase (*Streptomyces*)	-	2/2 (100%)	Leu-Pro-Pro-Pro-Pro-Pro	Leu-Pro-Pro-Pro↓Pro-Pro	[13]
M12 * [41/109]	M12.164	Lebetase	Q98995/MER0002591	6/14 (43%)	Bradykinin	Peptide-Pro↓Phe-Arg	[14]
	M12.338	BmooMPalpha-I (*Bothrops* sp.)	P85314/MER0104668	21/36 (58%)	Bradykinin	Peptide-Pro↓Phe-Arg	[15]
M13 * [11/13]	M13.005	Oligopeptidase O3 (PepO)	MER0001059	3/7 (43%)	Beta-casein	Peptide-Pro↓Val/Ile-Peptide	[16]
	M13.010	Oligopeptidase O2 (PepO)	O52071/MER0004645	4/4 (100%)	Beta-casein	Peptide-Pro↓Val/Ile-Peptide	[17]
M34 [3/3]	M34.002	Pro-Pro endopeptidase 1 (*Clostridium difficile*-type)	MER0494994	13/13 (100%)	Putative adhesin	Peptide-Asn-Pro↓Pro-Peptide	[18]
	M34.003	Pro-Pro endopeptidase 2	MER0328914	3/3 (100%)	Putative s-layer protein	Peptide-Pro↓Pro-Peptide	[19]
M64 [1/1]	M64.001	IgA peptidase (*Clostridium ramosum*-type)	MER0016067	2/2 (100%)	Immunoglobulin IgA1	Peptide-Pro↓Val-Peptide	[20]
M72 [2/2]	M72.002	CpaA g.p. (*Acinetobacter baumannii*)	MER1365492	2/2 (100%)	Coagulation factor XII	Peptide-Pro↓Thr-Peptide	[21]
S6 [4/8]	S06.001	IgA1-specific serine peptidase (*Neisseria*-type)	MER0000278	6/6 (100%)	Immunoglobulin IgA1	Peptide-Pro↓Xaa-Peptide	[22]
	S06.007	IgA1-specific serine peptidase type 1 (*Haemophilus* sp.)	MER0001759	3/3 (100%)	IgA1 chain C region	Peptide-Pro↓Thr-Peptide	[23]
S9 * [21/32]	S09.001	Prolyl oligopeptidase	P23687/MER0000392	50/59 (85%)	Alpha/beta-gliadin MM1	Peptide-Pro↓Xaa-Peptide	[24]
	S09.002	Prolyl oligopeptidase homologue (*Pyrococcus*-type)	MER0000398	2/3 (67%)	Z-Gly-Pro-NHPhNO2	Z-Gly-Pro↓NHPhNO2	[25]
	S09.003	Dipeptidyl-peptidase IV (eukaryote)	P27487/MER0000401	21/29 (72%)	C-X-C motif chemokine 10	Val-Pro↓Leu-Peptide	[26]
	S09.006	Dipeptidyl aminopeptidase B (fungus)	P18962/MER0000405	5/7 (71%)	Alanyl/prolyl bond	Xaa-Ala/Pro↓Xaa	[27]
	S09.007	Fibroblast activation protein alpha subunit	Q12884/MER0000399	6/6 (100%)	Alpha-2-antiplasmin	Peptide-Pro↓Asn/Leu-Peptide	[28]
	S09.008	Dipeptidyl peptidase 4 (*Aspergillus*-type)	MER0004504	6/8 (75%)	[Des-Arg]-bradykinin	Pro-Pro↓Gly-Peptide	[29]
	S09.009	Dipeptidyl-peptidase 4 (bacteria-type 1)	Q5NMM8/MER0041840	3/5 (60%)	Gly-Pro-NHNap	Gly-Pro↓NHNap	[30]
	S09.013	Dipeptidyl-peptidase 4 (bacteria-type 2)	Q47900/MER0001423	15/17 (88%)	Beta-casein	Tyr-Pro↓Phe-Peptide	[31]
	S09.017	Prolyl tripeptidyl peptidase	Q7MUW6/MER0005196	13/13 (100%)	Cystatin C	Ser-Ser-Pro↓Gly-Peptide	[32]
	S09.018	Dipeptidyl-peptidase 8	Q6V1X1/MER0013484	8/10 (80%)	C-X-C motif chemokine 10	Val-Pro↓Leu-Peptide	[33]
	S09.019	Dipeptidyl-peptidase 9	Q86TI2/MER0004923	13/15 (87%)	RU1 antigenic peptide	Val-Pro↓Tyr-Peptide	[34]
	S09.033	Prolyl oligopeptidase (zoomastigote)	Q4E132/MER0079308	-	-	-	[35]
	S09.036	Rv0457C peptidase (*Mycobacterium tuberculosis*)	MER0003229	2/2 (100%)	Suc-Gly-Pro-NHMec	Peptide-Pro↓NHMec	[36]
	S09.073	Xaa-Pro dipeptidylpeptidase	D7UPN5/MER0195666	19/19 (100%)	Ala-Pro-NHPhNO2	Xaa-Pro↓NHPhNO2	[37]
	S09.076	Prolyl oligopeptidase (*Myxococcus xanthus*)	Q9X5N2/MER0005694	2/2 (100%)	Suc-Ala-Pro-NHPhNO2	Peptide-Pro↓NHPhNO2	[38]
	S09.077	Prolyl oligopeptidase B (*Galerina marginata*)	H2E7Q8/MER0325901	3/3 (100%)	Alpha-amanitin proprotein 1	Peptide-Pro↓Ile/Trp-Peptide	[39]
	S09.UPA	Subfamily S9A unassigned peptidases	-	3/3 (100%)	Gly-Pro-NHPhNO2	Gly-Pro↓NHPhNO2	[40]
S15 [1/1]	S15.001	Xaa-Pro dipeptidyl-peptidase	Q02W78/MER0000443	8/10 (80%)	Gly-Pro-NHPhNO2	Gly-Pro↓NHPhNO2	[41]
S28 [2/3]	S28.001	Lysosomal Pro-Xaa carboxypeptidase	P42785/MER0000446	11/11 (100%)	Endothelin B receptor-like protein 2	Peptide-Pro↓Ala	[42]
	S28.004	Acid prolyl endopeptidase (*Aspergillus* sp.)	A2QR21/MER0093133	-	-	-	[43]
S33 [5/10]	S33.001	Prolyl aminopeptidase	P42786/MER0000431	4/6 (67%)	Consensus prolyl bond	Pro↓Xaa	[44]
	S33.004	Prolyl dipeptidase (*Lactobacillus*-type)	A8YWX2/MER0000437	17/18 (95%)	Pro-Gly	Pro↓Xaa	[45]
	S33.008	Prolyl aminopeptidase 2	P46547/MER0001367	9/10 (90%)	Pro-NHPhNO2	Pro↓NHPhNO2	[46]
S37 [1/1]	S37.001	PS-10 peptidase	Q54408/MER0001350	3/3 (100%)	Transglutaminase precursor	Pepetide-Pro↓Asp-Peptide	[47]
U9G [3/20]	U9G.029	Prolyl endopeptidase (*Spinacia*)	-	3/3 (100%)	Oxygen-evolving enhancer protein 3, chloroplastic	Peptide-Pro↓Ile/Leu-Peptide	[48]
U74 [1/1]	U74.001	Neprosin	-	51/110 (46%)	Alpha/beta-gliadin MM1	Peptide-Pro↓Xaa-Peptide	[49]

↓: cleavage site; -: the linked amino acids; NA: not available; Xaa: any amino acid. **^†^** Specificity is based on the MEROPS specificity matrix showing the proportion of proline residue in the P1 cleavage site of all substrate cleavage records. Numbers in square brackets indicate the proportion of subfamily with records of post-proline cleavage (https://www.ebi.ac.uk/merops/cgi-bin/specsearch.pl) (accessed on 15 March 2022) with query XXX-XXX-XXX-PRO|XXX-XXX-XXX-XXX. * Overrepresentation (*p* < 0.001) is based on the hypergeometric test.

**Table 2 plants-11-01330-t002:** The post-proline cleaving enzymes that are reported in the different species discussed in this review.

Taxa	Species (Common Name)	UniProt/PDB ID	MEROPS	Enzyme	Reference
Virus	Strawberry Mottle Virus	-/-	G03.001	Glutamic peptidase	[9]
Archaea	*Pyrococcus furiosus*	Q51714/5T88	S09.002	POP	[50]
Bacteria	*Aeromonus punctata*	Q9X6R4/3IUM	S09.001	PEP	[51]
	*Flavobacterium meningosepticum*	P27195/*-*	S09.UPA	POP	[52]
	*Lactobacillus helveticus*	O52071/*-*	M13.010	PepO-3	[17]
	*Meiothermus ruber* H328	A0A7C3HT26/-	S09.UPA	PEP/POP	[40]
	*Mycobacterium tuberculosis*	O07178/-	S09.036	POP	[36]
	*Myxococcus xanthus*	Q9X5N2/2BKL	S09.076	POP	[38]
	*Porphyromonas gingivalis*	Q7MUW6/2D5L	S09.017	PTP	[53]
	*Serratia marcescens*	O32449/1QTR	S33.001	PAP	[54]
	*Sphingomonas capsulate*	Q9ZNM8/-	S9.UPA	POP	[38]
	*Stenotrophomonas maltophilia*	A0A0U5BDB7/-	S9.UPW	PEP	[55]
Fungi	*Aspergillus niger*	A2QR21/7WAB	S28.004	AN-PEP	[43]
	*Aspergillus oryzae*	A0A1S9DCM9/-	S28.004	PEP	[56]
	*Galerina marginata*	H2E7Q8/5N4C	S09.077	POP-B	[39]
Protozoa	*Trypanosoma brucei*	Q6HA27/-	S09.033	POP	[57]
	*Trypanosoma cruzi*	Q71MD6/-	S09.033	POP	[58]
Insects	*Drosophila melanogaster*	Q10714/2X91	M02.003	ANCE	[59]
	*Eurygaster integriceps* (sunn pest)	E1U339/-	S09.001	PEP	[6]
	*Haematobia irritans* (buffalo fly)	Q10715/*-*	M02.003	PDP	[10]
Animals	*Haliotis discus* (abalone)	A0A1X9T5X9/6JYM	S09.UPA	PEP	[60]
	*Homo sapiens*	P48147/3DDU	S09.001	POP	[61,62]
		P27487/1J2E	S09.003	DPP-IV	[63]
		P42785/3N2Z	S28.001	Lysosomal PRCP	[64]
		Q9BYF1/1R42	M02.006	ACE-2	[65]
	*Mus musculus* (mouse)	P42676/1I1I	M03.002	Neurolysin (POP)	[66]
		Q9QUR6/*-*	S09.001	POP	[67]
	*Bothrops* sp. (snake)	P85314/3GBO	M12.338	BmooMPalpha-I	[15]
	*Sus scrofa* (porcine)	P23687/1O6G	S09.001	POP	[68]
Plants	*Arabidopsis thaliana*	F4HSS5/-	S33.001	PAP	[64]
	*Coffea arabica* (coffee)	-/-	S09.001	POP	[69]
	*Daucus carota* (carrot)	-/-	S09.UPA	PEP	[70]
	*Nepenthes* sp. (pitcher plant)	A0A1V0DK55/-	U74.001	Neprosin	[71]
	*Secale cereale* × *Triticum turgidum* subsp. *durum* (triticale)	G9J616/-	S33.001	PAP	[72]
	*Spinacia oleracea* (spinach)	P12301/-	U9G.029	PEP	[73]
	*Vigna radiata* (mung bean)	-/-	S28.002	DPP-II	[74]

Note: ACE/ANCE: angiotensin-converting enzyme; DPP: dipeptidyl-peptidase; PAP: prolyl aminopeptidase; PDP: peptidyl-dipeptidase; PEP: prolyl endoproteinase; PepO: oligopeptidase O; POP: prolyl oligopeptidase; PRCP: Pro-Xaa carboxypeptidase; -: not available. Italic PDB ID indicates the crystal structure for the homology modeling in SWISS-MODEL.

## 2. Molecular Structure and Biochemistry

### 2.1. Structural Studies

Protein structures of PPCEs have been studied across different taxa including bacteria, fungi, animals, and plants (Table 2). In this section, we selected some of the well-studied structures of PPCEs to depict their similarities and differences in hope of gaining insights on their regulation and molecular mechanisms for further investigations. The protein structures found in the PDB database are mostly solved through X-ray crystallography. The first crystal structure for a prolyl oligopeptidase (POP) was solved in 1998 at 1.4 Å resolution to reveal the α/β hydrolase fold and β-propeller domains with a catalytic triad of Ser-His-Asp [68]. Subsequently, crystal structures were acquired for *Sphingomonas capsulate* (1.8 Å resolution) and *Myxococcus xanthus* (1.5 Å resolution) [38]. 

Based on the sequence alignments and structural models, POP proteins from the MEROPS S9 family have two conserved domains (Figure 1). The C-terminal catalytic α/β-hydrolase domain comprises an alpha/beta/alpha sandwich for protein cleaving, whereas the N-terminal β-propeller domain constitutes β-sheets that limit proteolysis to smaller substrates as a mechanism to avoid non-targeted digestion [38,51,75]. The electron microscopy of human PEP revealed the presence of a new side opening that was not observed in any of the crystallographic structures [75]. Two paths were identified using the CAVER algorithm, leading from the PEP active site to the outside solvent, one through the β-propeller and another one through a large side aperture into the catalytic domain [75].

PEP/POP from different species have different β-propeller sizes, thus conferring specificity to different sizes of substrates targeting proline residues that are usually resistant to protein cleavage by other peptidases [76,77]. For example, studies on *Arabidopsis thaliana*, *Homo sapiens*, and *Sus scrofa* (porcine) showed that each POP possesses a unique affinity toward different sizes of ligands despite high sequence and structural similarities [78]. Besides the pore size of β-propellers, the number of β-propeller blades also plays a role in the substrate specificity of PEP. For instance, the porcine POP (S9a, S09.001) has a seven-bladed β propeller structure that acts as a filtering gate to exclude large peptides from the active site [68], whereas the crystal structure of the human dipeptidyl-peptidase (DPP IV) (S9b, S09.003) shows a unique eight-bladed β-propeller (Figure 2). The irregular blade-1 on DPP-IV is hypothesized to allow substrate entry to the catalytic site without going through the gating filter compared to POP [63]. The difference in the β-propeller of POP and DPP-IV could have contributed to the substrate specificity of these two enzymes in which POP hydrolyses small peptides (<30 amino acids) at the C-terminal of proline residues, while DPP-IV cleaves dipeptides at the penultimate proline residue. As compared to the β-propeller domains, the α/β-hydrolase catalytic domain is mostly conserved. 

There is a recent report of the *Aspergillus niger* PEP (AN-PEP) structure solved through an x-ray crystallography [43]. AN-PEP belongs to the peptidase family S28 of Pro-Xaa carboxypeptidase, which is in the same SC clan with family S9 sharing the same catalytic site residues (Ser-Asp-His) and likely the same evolutionary origin. The AN-PEP structure consists of 17 α-helices, 10 310helices, and 10 β-strands. In contrast to the POP of peptidase family S9, the β-propeller domain is replaced by a helical SKS domain (Figure 3) that is stabilized by three disulfide bonds. The catalytic pocket of AN-PEP is located between the catalytic α/β hydrolase domain and the SKS domain. AN-PEP is capable of digesting large substrates, unlike POP with a substrate limit of less than 30 amino acids. The substrate specificity could be due to the differences in the catalytic pocket structure formed between the α/β hydrolase and SKS domains in AN-PEP, compared to the β-propeller domain of S9 PEP/POP.

Another family of PPCEs is the prolyl aminopeptidase (PAP) from the peptidase family S33, which can be found in apricot seeds [24] and cabbage leaves [79]. Prolyl aminopeptidase (S33.001) is an exopeptidase that catalyzes the removal of the proline residue at the N-terminal of a peptide [54]. Like peptidase families S9 and S28, the catalytic α/β hydrolase domain is present (Figure 3). However, it has a catalytic triad of Ser113, His296, and Asp 268 with a consensus sequence of GXSXG around the catalytic serine residue. The second unannotated helices domain comprises six α-helices that act as a cap to block the N-terminal of the pre-cleavage (P1) proline, thus explaining the exopeptidase catalytic mechanism. PAPs are categorized based on their functioning structures: 30–35 kDa monomers (S33.001) are found exclusively in bacteria, while 100–370 kDa multimers (S33.008) have been reported in bacteria, fungi, and plants. Interestingly, a biochemically characterized plant PAP from triticale is more like the monomeric PAPs than the multimeric form [77].

On the other hand, a protease from the carnivorous tropical pitcher plants known as neprosin with PPC activity was classified as an unknown peptidase family (U74). There is a recent neprosin structure–function analysis based on the AlphaFold2 modeling that suggests neprosin belongs to the glutamic peptidase family with two glutamic acid residues as the catalytic dyad [80]. Unlike the S9, S28, or S33 prolyl proteases, neprosin does not have an α/β hydrolase domain, SKS domain, or β-propeller domain. Instead, neprosin has two uncharacterized domains, namely the neprosin activation domain and the neprosin domain (Figure 3). The neprosin domain is proposed to be the catalytic domain due to the absence of the neprosin activation domain in the active enzyme either in the native or recombinant neprosins [49,81]. Intriguingly, the neprosin domain comprises predominantly β-sheets, which form two antiparallel six- and seven-stranded β-sheets with an overall β-sandwich structure. 

In comparison, all the PPCEs from family S9, S28, S33, and neprosin have a two-domains structure. The α/β hydrolase domain and neprosin domain act as the catalytic domain, whereas the second domain (SKS, β-propeller, helical, and neprosin activation domains) appears to be involved in other functions such as substrate size limitation in S28 and S33 PEP, potential protein–protein interactions for S9 POP [82], and enzyme maturation/activation in the case of neprosin. Interestingly, two-domain structures are observed in these different families of PPCEs from diverse taxa, despite a lack of homology in amino acid sequences. This suggests a convergent functional and structural evolution.

PPC activity has also been reported in a few members of zinc metallopeptidase families (M2, M3, M9D, M12, M13, M34, M64, and M72). Peptidyl-dipeptidase ANCE (M02.003), angiotensin-converting enzyme-2, ACE2 (M02.006), and neurolysin (M03.002) comprise two domains annotated as zinc metallopeptidase domain I and domain II [65,66,83]. An active channel with a bound zinc ion is located in between the two active helices domains connected by small secondary structures (Figure 4). One of the structural differences between ACE2 and neurolysin is the number of the strand of the β-sheet on domain II, where ACE2 has a three-stranded β-sheet, while neurolysin has a five-stranded β-sheet (Figure 4). Furthermore, the more flexible loop structures on neurolysin could have contributed to its wide substrate specificity [66]. 

Metallopeptidase M12 BmooMPalpha-I (M12.338, 3gbo) with a molecular mass of 22.6 kDa has conserved features of P-I class snake venom metallopeptidases, namely the five-stranded β-sheet, four long helices, and a short α-helix near N-terminal stabilized by three disulfide bridges [15]. The zinc ion is bound to three histidine residues: His140, His144, and His150 (Figure 5). Additionally, Pro-Pro endopeptidase 1, PPEP-1 (M34.002, 5a0p), and Pro-Pro endopeptidase 2, PPEP-2 (M34.003, 6fpc) shared a very similar structure with an α/β N-terminal domain (NTD) comprising twisted four-stranded β-sheet and three α-helices (α1-α3), and an α C-terminal domain (CTD) with four α-helices (α5-α8) [19,84]. The active site of Pro-Pro endopeptidase 2 is located at the α4-helix that separates NTD and CTD domains. A zinc ion is bound to the catalytic base Glu138, two histidine residues (His137 and His141) from α4-helix, Tyr174 from α5/α6-helices, and Glu181 from α6-helix (Figure 5). The four amino acid substrate loops (S-loops) in PPEP-1 (GGST) and PPEP-2 (SERV) are shown to contribute to the difference in their substrate specificities [19]. Therefore, peptidases with proline-cleaving activity are not confined to the serine peptidase family alone. The advent of accurate ab initio protein structure prediction from sequences will reveal more peptidases conferring PPC activity via different catalytic mechanisms with different structures and provide a further understanding of the structure-function of the PPCEs. 

### 2.2. Molecular Mechanisms of PPCEs

Typically, PEP/POP from the clan SC of family S9 comprises a catalytic triad with the following order, Ser-Asp-His with the catalytic Ser lies in between the Gly-X-Ser-X-Gly motif [68]. However, there are also “nonclassical” serine proteases such as the Ser/His/Glu, Ser/His/His, and Ser/Glu/Asp triads, Ser/Lys and Ser/His dyads, and even peptidases with a single serine catalyst [85]. For S9 PEP, two major hypotheses on substrate binding mechanism had been proposed. The initial hypothesis is that the substrate enters the filtering gate and the central channel formed by the β-propeller structure before reaching the active site. This theory is supported by *Sphingomonas capsulate* PEP, as it failed to digest the 33-mer substrate (LQLQPFPQPQLPYPQPQLPYPQPQLPYPQPQPF) effectively with a diameter larger than the 4 Å channel formed by the β-propeller domain. However, this theory cannot explain why S9 PEP from *Myxococcus xanthus* and *Flavobacterium meningosepticum* can digest the 33-mer substrate in the same study. Later, it is proposed that substrate binding with the β-propeller domain induces conformation changes, causing the domain to move and expose the active site. This theory is supported by the open state of *Sphingomonas capsulate* PEP structure, where an opening of 30 Å is observed between the α/β hydrolase and β-propeller domain [38].

Another study obtained an open state of *Aeromonus punctata* PEP and proposed an induced-fit mechanism as the conformational changes in the PEP can be induced by adding substrate and inhibitor [51]. The open state is achieved when the substrate H2H3 is added, while a closed state occurs after the binding of the Z-prolyl-prolinal (ZPP) inhibitor with PEP (Figure 6). HP35 substrate with over 30 amino acids can enter the active site but is not cleaved, as the catalytic residue is not in an active conformation in the open state, whereas the catalytic pocket with the active catalytic triad conformation in a close state is too small for HP35. Recently, a molecular simulation of *Pyrococcus furiosus* POP (S09.002) shows that the conformational change is modulated by the “latching loop” mechanism and is essential for the catalysis mechanism of POP, especially for the loop containing catalytic histidine residue (H592) [50]. The conformation of the histidine loop during the shifting of open to close state allows H592 to move closer to the active site for the catalytic triad formation. Chloride ion binding is required for the loop movement with subsequent activation of the POP peptidase catalysis.

For peptidase family S28, it is suggested that the difference in the catalytic pocket located between α/β hydrolase and the SKS domain could confer substrate specificity based on the recent AN-PEP crystal structure [43]. The catalytic pocket formed in AN-PEP is wide-open compared to the lysosomal Pro-Xaa carboxypeptidase (PRCP) (S28.001) and dipeptidyl peptidase II, DPP7 (S28.002) with a volume-limited catalytic pocket that will affect the recognition of substrates with different sizes [64]. However, Glu88, Pro205, Trp374, and Trp460 residues are conserved across PEP, PRCP, and DPP7 despite differences in the volumes of catalytic pocket.

Different from S9 and S28 families, the S33 family has exopeptidase properties whereby it cleaves the proline residues at the N-terminal of a peptide. Prolyl aminopeptidase (PAP) from *Serratia marcescens* (S33.001) has a similar α/β hydrolase domain and another smaller helical domain made up of six helices, which are smaller than the SKS domain on S28. The smaller domain of prolyl aminopeptidase is believed to contribute to the exopeptidase specificity of S33 prolyl protease as the specificity hydrophobic pocket is formed on the helical domain [54]. Two mutagenesis studies on prolyl aminopeptidase from *Serratia marcescens* concur that residues Phe139, Tyr149, and Glu204 play an important role in the substrate recognition [86,87]. Phe139, Tyr149, and Phe236 in the hydrophobic pocket together with Glu204 and Glu232 are involved in electrostatic interactions. The proline recognition is completed by four electrostatic interactions and the insertion of substrate into the hydrophobic pocket of prolyl aminopeptidase [86]. Substrate acetylation could help in the orientation of the substrate in the active cleft for efficient hydrolysis [88]. The catalytic mechanism of peptidase family S28 and S33 could be like that of S9 prolyl endopeptidase, as they shared a similar classical serine catalytic triad of Ser-Asp-His.

Other than the serine peptidases, a glutamic peptidase was reported with PPC activity and a catalytic dyad comprising two catalytic glutamic acid residues [9]. In silico analysis of neprosin found a high structural similarity to strawberry mottle virus glutamic peptidase (SMoV peptidase, G03.001) with an overlapping catalytic dyad [80]. This shed light on the possible catalytic mechanism of neprosin. The mature neprosin has a putative accessible active cleft formed by the β-sandwich structure for substrate binding. The catalytic mechanism of glutamic peptidase G3 could be like that of the Gln and Glu catalytic dyad in the G1 family because the main catalytic Glu residue of glutamic peptidase G1 is found to be conserved in the G3 family [89].

### 2.3. Biochemical Studies of PPCEs

The native (Table 3) and recombinant proteins (Table 4) of PPCEs have been reported for different species, mainly microbes and animals. For the assays of PPC activity, Z-Gly-Pro-*p*NA (ZGPpNA) and other chromogenic substrates are commonly used [90]. PPCEs specifically cleave at the carboxyl-end of proline residue of the chromogenic substrates, releasing chromogenic *p*NA (*p*-nitroanilide) or βNA (β-naphthylamide), which can be detected spectrophotometrically at a wavelength of 410 nm and 520 nm, respectively. Due to their specificity, these chromogenic substrates are often used in the functional characterization and activity measurement of PPCEs. Furthermore, proline-rich gliadin is also a common substrate to investigate the capability of certain PPCEs in gluten detoxification. The fragments of gliadin hydrolysates can be detected through ELISA, Western blot, or immunoblot with specific antibodies. To ascertain the protein cleavage sites, a liquid chromatography–mass spectrometry (LC-MS) approach can be taken to identify the digested peptide sequences. 

Specific amino acid preferences of targeted substrate sequences are curated in the MEROPS database with a collection of cleavage sites (Table 1). The substrate specificity will influence the molecular interactions between the protease and substrate at the active site of the protein with a role in biological functions [91]. Experiments to identify the substrate specificity of PPCEs utilize LC-MS [81] and fluorogenic substrates [92]. One study used the spectrofluorometer to determine the fluorometric Z-Gly-Pro- *p*NA peptide at the excitation (340 nm) and emission wavelengths (410 nm) [92]. In this study, the POP enzyme was shown to form strong interactions with no more than six residues from positions P4, P3, and P2 (N-side) and P1′ and P2′ (C-side) regardless of the substrate length [92]. 

PPCEs can hydrolyze proteins with different preferences of amino acids adjacent to the proline residues. For example, the recombinant PEP from a Gram-negative thermophile, *Meiothermus ruber* H328, shows a stringent preference of residues next to proline but a greater flexibility in preferences of the second and third residues near proline [40]. In a study using 3375 synthetic peptides as substrates of internally quenched fluorogenic probes (IQFPs), about 74% showed high favorability towards proline at the Xaa position [93]. Both cationic and anionic charged residues such as Arg/Lys/Asp/Glu are unfavorable at the subsequent position after proline. The anionic residues are more tolerated in position P2 compared to P3 with strong preferences toward Leu/Ile/Arg/His and low preferences towards Asp/Glu at P3 positions [94]. Therefore, PPCEs have a certain degree of specificity towards their substrates based on their amino acid sequences. 

Studies on both the native and recombinant PPCEs, mainly PEP/POP, showed similar optimum temperatures with pH 4–6, except for *Eurygaster integriceps*, *Haliotis discus*, *Sus sucrofa*, *Porphyromonas gingivalis*, and *Nepenthes* × *ventrata* (Table 3 and Table 4). Most of the native protein studies were based on *Aspergillus niger* with pH 4–5 and 37 °C. Nearly all recombinant protein characterization studies used *Escherichia coli* as a cloning and expression host apart from *Pichia pastoris* and wheat (Table 4). Some PPCEs exhibited different preferences for different substrate lengths and specificity with a broad range of optimum temperatures and pH levels. For instance, a POP (S09.001) cleaves after the C-terminal proline residue in peptide substrates with less than 30 amino acids. 

In addition to the PPC activity that cleaves at two conserved proline residues of α-amanitin pro-peptide, POP-B from *Galerina marginata* (GmPOPB, S09.077) has a unique transpeptidation activity that catalyzes macrocyclization of a 25-mer peptide to form a monocyclic octapeptide [39]. A PEP from *Aspergillus oryzae* (S28.004) exhibits a similar PPC activity at an optimal pH of 2.5 with the capability of digesting much larger substrates such as intact casein [95]. Like S28 PEP, neprosin with a lower molecular mass (~30 kDa) shows PPC activity at pH 2.5 but with no limitation of substrate size [81]. Prolyl aminopeptidase (S33.001) from family S33 has a proline-specific exopeptidase activity that cleaves proline at the N-terminal of a peptide [54]. All PPCEs showed the ability to cleave proline-rich substrates. These studies show the potential of PPCEs from different species for industrial applications with various conditions of pH and temperatures.

**Table 3 plants-11-01330-t003:** The optimum conditions of selected native PPCEs in substrate degradation.

Source	Enzyme	Substrate *	pH	T (°C)	Method to Detect Digested Substrate *	Reference
**Fungus**						
*Aspergillus niger*	PEP	Beer hordein	-	-	ELISA RIDASCREEN^®^ Gliadin competitive	[96]
			4–5	-	Gluten ELISA assay	[97]
			-	14	Antibody-based competitive enzyme-linked immunosorbent assay	[98]
			4	37	Gluten ELISA assay	[99]
			4.6	37	Western blot, monoclonal antibody-based competition assays	[100]
			4–5	37	SDS-PAGE, Western blot, ELISA, HPLC, MS,	[101]
		Wheat flour, gliadin	4	40	ELISA, immunoblot (anti-PEP-I, anti-PEP-II, and anti-HMW-GS antibodies), RP-HPLC	[7]
		N-glycosidase	4–4.5	37	MALDI-TOF MS	[102]
		ZGPpNA	4	37	Turbidity (protein-polyphenol haze)	[103]
			4–5	40–50	UV-Vis 410 nm	[104]
**Insect**						
*Eurygaster integriceps* (sunn pest)	PEP	ZGPpNA	8 and 10	24–34	UV-Vis 410 nm	[6]
**Animal**						
*Mus musculus*	POP	Suc-Gly-Pro-AMC	4.2	30	Fluorescence plate reader	[67]
**Plants**						
*Nepenthes* × *ventrata*	Neprosin	α-gliadin	2.5	37	SDS-PAGE, turbidity monitored at 595 nm, MS/MS	[49]
*Spinacia oleracea* PSII membranes	PEP	Co-purified 23 kDa and 18 kDa proteins	6	37	Densitogram	[73]
*Vigna radiata*	DPP-II	-4mβNA and -βNA dipeptides	7.5	37	UV-Vis 520 nm	[74]

* -4mβNA: -4-methoxy-β-naphthylamide; -βNA: -beta-naphthylamide; ELISA: enzyme-linked immunosorbent assay; MS: mass spectrometry; MS/MS: tandem MS; RP-HPLC: reversed-phase high-performance liquid chromatography; NA: not available; SDS-PAGE: sodium dodecyl sulfate-polyacrylamide gel electrophoresis; ZGPpNA: benzyloxycarbonyl-Gly-Pro-p-nitroanalide.

**Table 4 plants-11-01330-t004:** The optimum conditions of selected recombinant PPCEs in substrate degradation.

Source	Enzyme	Expression Host	Expression	Purification	Substrate	pH	T (°C)	Assay Detection	Reference
**Fungi**									
*Aspergillus niger*	AN-PEP	*Pischia pastoris* GS115/pPIC9K	28 °C/96 h	Ni-affinity chromatography	ZGPpNA	4–5	35–40	UV-Vis 410 nm	[95]
							35	LC-MS/MS	[105]
*Aspergillus oryzae*	Prolyl aminopeptidase	*Escherichia coli* BL21/pET-28a(-)	25 °C/12 h	Ni-affinity chromatography	L-Pro-pNA	6.5–7.5	50	-	[106]
**Bacteria**									
*Flavobacterium meningosepticum*	Fm-POP	Wheat/pUC57	-	-	Gluten peptide	6	37	HPLC, ELISA	[52]
		*E. coli* DH5α/pUC57 + Fmen	37 °C/14 h	Geneclean III kit	ZGPpNA, gliadin, glutenin	-	Up to 90	UV-Vis	[107]
*Meiothermus ruber* H328	MrPEP	*E. coli* BL21(DE3)/pET-28b	37 °C/14 h	DEAE-cellulose and Phenyl-FF chromatography	ZGPpNA, suc-Ala-Pro-*p*NA, Gly-Pro-*p*NA, Pro-*p*NA	9	60	UV-Vis, FRETS-25Xaa libraries, LC-MS	[40]
*Myxococcus xanthus*	PEP	*E. coli* BL21(DE3)/pET-28b	22 °C/16 h	Ni-affinity chromatography	Collagen peptide	6–7	36–37	UV-HPLC-MS/MS	[24]
*Porphyromonas gingivalis*	Prolyl tripeptidyl aminopeptidase	*E. coli* M15/pQE30-ptpA	30 °C/18 h	Butyl-sepharose column	Fluorogenic substrate	7.5	30	UV-Vis	[53]
*Sphaerobacter thermophiles*	PEP	*E. coli* DE3/pET-15b	37 °C/24 h	Ni-NTA resin spin column	ZGPpNA, gluten peptide	6.6	63	Enzymatic assay, MS	[108]
*Sphingomonas capsulate*	PEP	*E. coli* BL21(DE3)/pET-28b	22 °C/16 h	DEAE and Blue sepharose chromatography	Chromogenic Gluten peptide	6–7	36–37	UV-HPLC-MS/MS	[24]
**Insect**									
*Eurygaster integriceps*	PEP	*E. coli* BL21(DE3)/pNYCO	28 °C/16 h	Ni-affinity chromatography	ZGPpNA, wheat gluten	7.5	37	UV-Vis	[109]
*Haematobia irritans exigua*	ANCE	-	-	Lectin-affinity and ion-exchange chromatography	Angiotensin 1, cholecystokinin-8	7.5	37	HPLC	[10]
**Animals**									
*Haliotis discus* (abalone)	PEP	*E. coli* BL21(DE3)	18 °C/15 h	Ni-affinity chromatography	Collagen peptide	6	20	HPLC-MS	[38]
*Sus sucrofa*	POP	*E. coli* Top10/pBAD	37 °C/4–6 h	DEAE and Blue sepharose chromatography	ZGPpNA -AMC	5–8	30	UV-Vis	[110]
**Plant**									
*Nepenthes × ventrata*	Neprosin	*E. coli* Arctic Express/pET-28a(+)	16 °C/16 h	Ni-affinity chromatography	HeLa cell protein	2.5	37–50	MS	[81]
					Histone	2.5	37	LC-MS/MS	[111]

Note: AMC: 7-amido-4-methylcoumarin; DEAE: diethylaminoethyl; ELISA: enzyme-linked immunosorbent assay; Hb: hemoglobin; HPLC: high-performance liquid chromatography; L-Pro-pNA: L-arginyl-L-proline 4-nitroanilide; MS: mass spectrometry; Ni-NTA: Ni^2+^-nitrilotriacetate; UV-Vis: ultraviolet-visible spectroscopy; ZGPpNA: Z-Gly-Pro-4-nitroanilide; -: not available.

## 3. Biological Functions of PPCEs

### 3.1. Microbes and Protozoa

Many PPCEs that have been purified and characterized are from microbes. The strawberry mottle virus glutamic peptidase unit (G03.001) can cleave at P↓AFP sites of the coat protein domain in the polyprotein and is hypothesized to be involved in the regulatory step of controlling viral RNA encapsidation [9]. Proline iminopeptidase or proline aminopeptidase (PAP) from family S33, including *Flavobacterium meningosepticum* and *Serratia marcescens* (S33.001), could be involved in pathogenicity by breaking down peptides rich in proline and hydroxyproline such as collagen [88]. Rv0457c peptidase (S09.036) from the S9 family is a POP found in *Mycobacterium tuberculosis* that can stimulate the secretion of proinflammatory cytokines by peritoneal macrophage and lead to inflammatory response during *M. tuberculosis* infection [36]. Besides, POP-B of *Galerina marginata* (S09.077) from the same family is required for the maturation of α-amanitin, a toxin that is responsible for the fatal mushroom poisoning [39]. There is also a proline-Xaa carboxypeptidase (AoS28D, S28.004) from *Aspergillus* sp. with different biochemical properties than the other paralogs from gene duplication, which functions together with other secreted aminopeptidases and carboxypeptidases to digest large peptides for uptake [112]. 

On the other hand, the POP zoomastigote (S09.033) from the unicellular parasitic flagellate protozoa *Trypanosoma cruzi* and *T. brucei* are involved in the pathophysiology of Chagas disease and human African trypanosomiasis, respectively [58]. The *Trypanosoma* spp. attached to the extracellular matrix (ECM) releases POPs that degrade collagen. The entry of *Trypanosoma* spp. with degraded collagen peptides signal host leukocytes to release matrix metalloproteases (MMPs) that further break down the ECM barrier. Ultimately, this allows the parasite to break through protective barriers such as ECM or the blood–brain barrier in causing diseases. These examples demonstrate that microbial and protozoa PPCEs have diverse biological roles, many of which are important in pathogenicity.

### 3.2. Animals

PPCEs have been associated with diverse physiological roles in animals, especially humans. In mice, POP (S09.001) was highly expressed in the brain, kidney, testis, and thymus, with the kidney having the least enzymatic activity [67]. Moreover, POP was also found in the mouse peripheral tissues and placental extract, showing that it might have important functions in these regions. An immunohistochemistry study has unveiled high POP densities in the caudate nucleus and putamen, hippocampus, and cortex [113]. POP activities are widespread but cell-specific in peripheral tissues, especially the secreting epithelial cells in the stomach, breast ducts, kidney, pancreas, and reproductive system [62]. Like mice, a high amount of POP is detected in the human testis, particularly in nuclei of primary spermatocytes and spermatids, oocytes, and primary follicles. Furthermore, POP is found to be elevated especially in malignant tumors [62]. Additionally, proinflammatory POPs are involved in the pathology of lung diseases including chronic obstructive pulmonary disease (COPD) and COVID-19 [114,115]. Proinflammatory POPs together with matrix metalloproteinases (MMPs) can generate matrikine proline-glycine-proline (PGP) by breaking down the proline-rich collagen of the ECM. The end-product of POP cleavage, prolyl-glycyl-proline (PGP), is then acetylated into AcPGP, which acts as a chemoattractant for neutrophils recruitment and continuously drive the cycle of neutrophilic inflammation [116,117].

PPCEs appear to play an important role in memory and learning as well as cognitive and behavioral development. For example, alterations of a POP and dipeptidyl-peptidase IV (DPP-IV, S09.003) levels in different brain parts were observed in mice of different genders administered with DPP-IV inhibitor [118]. DPP-IV was also shown to be involved in the regulation of glucose homeostasis in the body through the cleavage of glucose-dependent insulinotropic polypeptide-1 (GIP-1), an incretin hormone that is secreted upon nutrient uptake [119]. The administration of DPP-4 inhibitors in the clinical treatment of type 2 diabetes significantly enhances insulin levels [119].

Other than the localization of POP in the brain, the distribution of POP in the human brain resembled that of inositol 1,4,5-triphosphate (IP3) receptors, which supports the theory that POP plays a role in movement regulation, cognition, and IP3 signaling [113]. High POP activity in the human cortex suggests a role in memory, cognition, and learning. The POP is known to degrade a neuropeptide (substance P) with a role in promoting memory [120]. The inhibition of the POP enhanced IP3 concentration as a result of increased substance P, which led to cognitive enhancement in rats [121]. 

Altered POP activities were found in patients suffering from cognitive diseases and other health implications. Increased POP activity is associated with inflammation [122] as well as mania and schizophrenia [123], whereas decreased POP activity is related to depression [124], multiple sclerosis [125], anorexia, bulimia nervosa [126], Alzheimer’s disease, Lewy body dementia, Parkinson’s disease, and Huntington’s disease [127]. Moreover, POP and DPP-IV have been associated with aggressive behaviors in normal and autistic adolescents [128]. 

On the other hand, peptidyl-dipeptidase ANCE (M02.003) is involved in the renin-angiotensin-aldosterone system (RAAS) that regulates blood pressure by hydrolyzing angiotensin I into angiotensin II (vasoconstrictor) and its physiological substrate bradykinin, which is a vasodilator [10]. Moreover, angiotensin-converting enzyme-2 (ACE-2, M02.006) is a regulator of heart function. Based on *ace2* gene knock-out experiments, disruption in mice causes severely impaired heart function whereas the disruption of ace2 homolog in *Drosophila* embryo resulted in a serious defect in the heart morphogenesis [129]. Additionally, ACE2 is a functional receptor for SARS coronavirus [130], as shown in the crystal structure (PDB ID: 6M0J) of ACE2 bound to the SARS-CoV-2 spike with their receptor-binding domains [131]. 

Lastly, neurolysin with PPC activity is widely distributed in the mammalian tissues and can be found in different subcellular locations, in cytosolic, membrane-bound forms, or with mitochondrial targeting sequences [132]. It is a neuropeptidase that cleaves short neuropeptides such as neurotensin, 13-residues neuropeptides that involve in the homeostasis of blood pressure, body temperature, gastrointestinal movement, and the release of brain hormones (dopamine, luteinizing hormone, and prolactin). Although there is no concrete evidence, neurolysin is hypothesized to be involved in the pathology of human diseases such as epilepsy, angiogenesis, tumor growth, sepsis, stroke, and metabolic disorder, together with thimet oligopeptidase (THOP1) and a POP [132].

### 3.3. Plants

The first plant PEP was isolated from carrot (*Daucus carota*) from the screening of over 40 vegetables and fruits, including spinach, carrot, pumpkin, soybean, green onion, and parsley for PEP activity [70]. Most of them showed low activity, with spinach and carrot showing the highest PEP activity (1.11 unit/g and 0.81 unit/g, respectively). Carrot PEP was further analyzed since its extraction is easier, while the enzyme activity of spinach was more varied. The enzyme was shown to be active at pH 7.3 and stable between pH 6 and 8.5. The enzyme showed activity towards Z-Gly-Pro-β-naphthylamide as the substrate, which is like PEP found in mammals and microbes. Further research showed that the enzyme purified from the carrot has more similarities to mammals than microbes [70]. Besides that, PEP extracted from spinach can recognize the scissile prolyl bond to cleave the 18 kDa protein of photosystem II, indicating a possible role in chloroplasts and photosynthesis [73]. 

POP genes (*CaPEP1*, *CaPEP2*, and *CaPEP3*) were found to be differentially expressed in two closely related *Coffea arabica* varieties, cultivars “Tall Mokka” and “Typica” [69]. These genes are responsible for the lateral shoot branching and branched phenotype. Additionally, dipeptidyl-peptidase (DPP) II (S28.002) is shown to be involved in the protein mobilization during seed germination to provide nutrients for the growth of embryonic plants. A significant correlation between DPP-II activity, protein content, and free amino acid content during seed germination of *Vigna radiata* suggests that DPP-II breakdown protein storage in seed into free amino acid during the germination period [74]. TsPAP1 gene encoding a prolyl aminopeptidase (S33.001) was upregulated in the shoots of triticale under harsh environments such as dry and salty conditions, as was the presence of cadmium and aluminum ions in nutrient media [72]. Furthermore, the overexpression of TsPAP1 in transgenic *Arabidopsis thaliana* resulted in a faster flowering process and the production of more siliques [77]. These findings suggest that PPCEs play important roles in different plant growth and developmental stages. These PPCEs could be targets for future crop improvement [78]. 

On the other hand, omics studies have been conducted to discover digestive proteins that contribute to the botanical carnivory [133]. Enzymes in the pitcher fluids of the tropical carnivorous *Nepenthes* species were found mainly comprising aspartic protease nepenthesins and neprosins [134,135], which suggests the importance of proteases in the adaptation and diversification of *Nepenthes* species. Furthermore, PPC activity has been shown in the native and recombinant neprosins to degrade gliadins from wheat with high potency [49,81]. 

## 4. Biotechnological Applications

### 4.1. Industrial Applications

PPCEs have many uses in industry. For example, PEP enzymes are useful for the gluten-free food processing industry for people with gluten intolerance, such as celiac disease. Celiac disease occurs in individuals with HLA-DQ2 or HLA-DQ8 genetic predisposition, where the ingestion of gluten can trigger a T-cell immune response in the small intestine [136]. Gluten is a common protein that can be found in cereals such as barley and wheat, which are proline-rich and resistant to digestion by pancreatic or gastric enzymes. High proteolytic resistance gluten peptides such as alpha-gliadin 33-mer and gamma-gliadin 26-mer peptides can bind to the HLA-DQ2 or HLA-DQ8 antigens. These peptides are presented to the pro-inflammatory T-cells, which can cause inflammation and damage to villi on the small intestine wall, leading to various complications [137]. The best treatment for celiac disease is a strict gluten-free diet to relieve symptoms, the reversal of malabsorption, and the restoration of villi. A study investigating five gluten-degrading enzyme supplements on the market showed that amylases, galactosidases, proteases, and/or subtilisin cannot eliminate the nine immunogenic epitopes of 26-mer and 33-mer [138]. In contrast, AN-PEP degrades all nine epitopes effectively at the pH range of the stomach at a much lower dose [7]. AN-PEP (S28.004) has been tested against three different types of wheat flours, *Triticum aestivum* (HD-2851, NIAW-917), *Triticum durum* (UAS-428), and *Triticum dicoccum* (DDK-1025) for proteolytic degradation and shown to have 99.95% less gluten content than the control pasta. The sensory evaluation test showed that the quality of the treated wheat flour is comparable to the control pasta, despite having a brownish color [7]. This demonstrates the potential of PEP to be used in oral supplements for gluten detoxification. Another study showed that the consumption of bread pre-digested with AN-PEP lowered the immunogenic gluten by approximately 40% [96]. Hence, AN-PEP pre-digestion is applicable in the bread production industry. 

Furthermore, to test the efficiency of AN-PEP gluten degradation in a randomized placebo-controlled crossover study, 18 celiac disease patients were given a porridge containing 0.5 g gluten together with two tablets either containing a high or low dose of AN-PEP, or a placebo. In a meal physiological setting, samples were then taken from the stomach and duodenal content to calculate the gluten content. The results showed over a 50% reduction in gluten content in both stomach and duodenum as compared to the placebo. Furthermore, gluten degradation was shown to be more apparent in the stomach compared to before it entered the duodenum, showing AN-PEP as a potent protease in degrading gluten content when provided in a meal setting [97].

Another study showed that AN-PEP degrades gluten in a dynamic system (TIM) that closely mimics the human gastrointestinal tract [100]. Gluten content measurements were taken from the stomach, duodenum, jejunum, and ileum compartments of individuals who ingest bread or standard fast food processed in the TIM system with and without the co-administration of AN-PEP. Based on monoclonal antibody-based competition assays, Western blot analysis, and T-cell proliferation assays, most gluten was degraded in the stomach before it reaches the duodenum compartment. Hence, this experiment also showed that AN-PEP degrades gluten in a system that mimics human in vivo digestion [100].

There are also efforts to utilize POP from *Sphingomonas capsulate* and *Myxococcus xanthus* as oral supplements for gluten degradation, but these proteases failed to detoxify gluten after irreversibly inactivated by pepsin and acidic pH in the stomach [101]. Instead, AN-PEP is more resistant to the acidic pH and pepsin in the stomach and has proven potent against gluten. However, reports also indicate that different types of meals need a different amount of AN-PEP for gluten detoxification. Acidic meals such as carbonated drinks enhance AN-PEP activity, while food proteins reduce the AN-PEP capacity for gluten detoxification, while not being affected by fats. Raw gluten is easier to be degraded by AN-PEP than baked gluten. Therefore, AN-PEP is not able to digest large portions of gluten; hence, it should not be used to replace an entirely gluten-free diet, but rather to support the digestion of low occasional gluten consumption [99]. On the other hand, the combination of native neprosin and nepenthesin proteins from the tropical pitcher plant *Nepenthes* × *ventrata* has demonstrated the capability to digest gliadin at a higher potency than AN-PEP [49]. This shows that neprosins could be a better alternative of AN-PEP for celiac disease enzyme therapy. 

Meanwhile, PEP can reduce the hordein content in beer that may interact with polyphenol causing cloudy precipitation (chill-haze) during storage [104]. One alternative to removing the polyphenols that interact with the proline-rich residues is by removing the proline-rich proteins that are intolerant for gluten-sensitive individuals. Metallopeptidase M13 oligopeptidase O2 (M13.010) expressed in *Lactobacillus helveticus* WSU19 with PPC activity can effectively debitter aged cheddar cheese. The oligopeptidase O2 was experimentally proven to cleave three sites after Pro196, Pro200, and Pro206 for the complete digestion of the bitter β-casein (f193–209) {Chen, 2003 #17}.

A study also showed that barley treated with AN-PEP for five days during germination can reduce beer hordein content by 46% compared to the control when tested using an R5 antibody-based competitive enzyme-linked immunosorbent assay [98]. In theory, a longer malting process will incur a higher cost, but the treatment of AN-PEP during the malting process does not affect the malting loss significantly [98]. Moreover, to improve the enzymatic activity of PEP, a study used immobilized proline-specific endoprotease (PSEP) or silica gel to prevent haze from forming inside the beer. Using the immobilized PSEP, it could retain up to 65% of its enzymatic activity as compared to the non-immobilized ones with better tolerance towards temperature and pH. Moreover, the enzyme can retain over 90% of its original activity even after six cycles. Hence, this provides an advantageous alternative for the beer brewing industry [104]. 

### 4.2. Therapeutic Agents

Due to the important functions of certain PEP/POP in human cognitive diseases (described in 3.2), understanding the mechanism of PPCEs is useful for the design of drugs or enzyme inhibitors as therapeutics. PEP inhibitors can be categorized broadly into natural and synthetic inhibitors. Natural PEP inhibitors can be found in traditional medicinal herbs, such as isoquinoline alkaloid berberine from *Berberis vulgaris*, flavonoid baicalein from *Scutellaria baicalensis Georgi*, and alkaloid californidine from *Eschscholzia californica* [139]. Berberine and its derivatives were found to be effective for bipolar disorder and patented as PEP/POP inhibitors in 2008 [140]. The current available synthetic PEP/POP inhibitors have been reviewed by Babkova et al. [139]. For example, cyanopyrrolidine-based compounds such as CbzMetPrdN can penetrate the blood–brain barrier in exerting beneficial anti-amnesic effects on mice. On the other hand, a study by Jalkanen et al. showed that POP inhibited by KYP-2047 inhibitor does not affect the level of neurotensin and substance P in the striatum extracellular space [141]. Notably, there is no consensus on the mechanism of POP in human cognitive diseases to date. Furthermore, there are multiple PPIs of POP such as the aggregation of α-synuclein [142], colocalization with α-tubulin [143], the inhibition of glyceraldehyde-3-phosphate dehydrogenase (GAPDH) mRNA [144], and interaction with growth-associated protein 43 (GAP43) [145].

Aside from PEP/POP, metallopeptidase M3 neurolysin (M03.002) with the ability to cleave multiple types of neuropeptides hinted at its potential as a single therapeutic target for multiple diseases. Hence, neurolysin and POP are currently being studied systematically as potential therapeutic targets to cure human brain diseases [146]. Moreover, metallopeptidase M2 such as peptidyl-dipeptidase ANCE (M02.003) and angiotensin-converting enzyme-2 (M02.006) are also therapeutic targets for human diseases. For instance, peptidyl-dipeptidase ANCE inhibitors were used in hypertension treatment [147]. Furthermore, a cell membrane chromatography study demonstrated that the third-generation antihistamine azelastine has an affinity toward ACE2 and provides antiviral properties by inhibiting the entry of pseudovirus in vitro [148]. Hence, azelastine could be a potential candidate for drug repurposing in the COVID-19 treatment. 

Other than human PPCEs, rational drug designs that target the POP of *Trypanosoma brucei* and *T. cruzi* are currently in progress to combat Chagas diseases and human African trypanosomiasis {Bastos, 2005 #58}. A recent study has shown that Tc80 POP from *T. cruzi* can be used as an antigen for the development of an effective vaccine against *T. cruzi* infection by reducing the parasite burden and tissue damage induced in the vaccinated mice [149]. The associations of PPCEs in many human diseases suggest that PPCEs could be potential biomarkers or therapeutic targets for diagnostic and treatment purposes. 

### 4.3. Proteomics

PPCEs with prolyl cleaving activity is different from other peptidases such as trypsin that is commonly used in the MS-based proteomics [150]. Conventional bottom-up MS uses trypsin that cleaves after lysine and arginine to produce peptide fragments. Although this cleavage pattern eases the interpretation of mass spectra, the high prevalence of lysine and arginine residues in histone proteins resulted in very small peptides and erodes post-translational modification (PTM) patterns that are sought after in histone mapping. The chemical derivatization of lysine with propionylation to prevent trypsin cleaving after lysine is the main approach for generating longer peptides. Newer middle-down proteomics approaches use alternative enzymes such as GluC, which cleaves the carboxyl-end of glutamic acids to produce a 50 amino-acid-long (aa) N-terminal tail for histone H, or AspN, which cleaves at the amino-end of asparagine residue to generate a 23 aa N-terminal tail for histone H4. Although they are more effective than the bottom-up approach using trypsin, the methods require different LC technology such as derivatization of unmodified lysine for reversed-phase liquid chromatography (RPLC) or weak cation exchange hydrophilic interaction liquid chromatography (WCX-HILIC) due to the low retention of histone H3. The derivatization complicates the routine of MS, while WCX-HILIC is technically challenging. 

PPCEs can aid in the generation of peptide fragments with a better grouping of PTM. This is due to the nature of PPCEs, which cleave after proline can disrupt protein secondary structures and are usually unmodified. However, not all PPCEs are suitable for the task due to their substrate size limitation and specificity. For instance, POPs are unable to cleave peptides larger than 30 residues. AN-PEP has a cleavage specificity outside proline and alanine while having broader substrate recognition. The activity of AN-PEP for MS is comparable to that of conventionally used pepsin with an equivalent number of peptides generated by AN-PEP at a lower concentration [151]. Moreover, the peptides generated are shorter than the peptides generated by pepsin, allowing better insight with higher resolution [151]. 

Neprosin with PPC activity was applied in the MS analysis of histone modifications due to its selective prolyl endoprotease nature as compared to the conventional trypsin-based methods to study the information of the histones H3 and H4 [111]. Neprosin without the substrate size limitation and higher proline specificity (with around 50% more cleavage of proline residues at position P1) are the most promising PPCEs for MS applications. Neprosin from *N.* × *ventrata* was used to digest 1251 HeLa cell proteins for a proteomics study [81], which showed that almost half of the result gives unique peptides compared to trypsin. This allows the detailed analysis of the protein PTM. Later, a simplified middle-down proteomic method utilizing neprosin enzyme for histone mapping was proposed, which can generate suitable peptide lengths for the retention of histone H3 and H4 tails, and histone tail analysis can be achieved in a single RPLC run. The authors also suggested the use of neprosin in gene-specific chromatin immunoprecipitation (ChIP) MS techniques, as this method does not require as much recovered protein sample as the method using alternative enzymes such as GluC and AspN. These studies demonstrated that neprosin can be exploited to expand proteomics tools. Mass spectral interpretation and fragment ion formation based on neprosin could be an interesting topic for future research. 

## 5. Protein Engineering

Studies to improve the activity of PEP enzymes have been performed through protein engineering. For example, PEP mutagenesis on residues F263A and E184G on the β-propeller domain of the novel *Stenotrophomonas maltophilia* prolyl endopeptidase (SmPEP) improves the enzymatic activity on synthetic peptides. The introduction of mutants on these specific residues weakens the inter-domain interaction of the enzymes to reduce the restriction of bigger substrates to enter the active site [55]. The increased specificity of the enzyme activity was recorded when random and site-directed saturation mutagenesis was performed on a prolyl aminopeptidase derived from *Aspergillus oryzae* [106]. Mutagenesis at C185V allowed a higher resistance towards high temperature, broader acidity range, and longer half-life compared to the wild type, making it a good candidate for many industrial sectors [106].

The amino acid mutation (Glu289Gly) of a human recombinant PEP prolongs the half-life of the enzyme in phosphate buffer at 37 °C to be used for in vivo Antibody-Directed Enzyme Prodrug Therapy (ADEPT) [152]. The production of a thermally stable PEP as part of ADEPT is necessary to improve the selectivity of the administrated prodrug towards the tumors without intoxicating other normal tissues [152]. Moreover, site-directed mutagenesis of POP from *Flavobacterium meningosepticum* (Fm-POP) that was expressed in the wheat endosperm showed the ability to reduce gluten content under gastrointestinal conditions with a higher ability to tolerate high temperatures over 37 °C up to 90 °C [52]. Gluten peptides can be digested up to 67% using Fm-POP in combination with barley endoprotease B2 (EP-HvB2) [107]. Hence, the modifications of these enzymes allow them to function under harsh conditions without affecting their enzyme efficiencies.

Moreover, the co-administration of cysteine endoprotease EP-B2 with improved mutagenesis variants of PEP from *Sphingomonas capsulate* showed the improvement in enzyme activity when simulated under gastric conditions [153]. The cysteine endoprotease breaks down the main gluten component in bread, while PEP degrades the residual peptide products generated from the earlier digestion [153]. The PEP variants produced from this study were able to detoxify gluten within a shorter time (10 min) compared to the wild-type PEP (60 min) in Gass et al. [154]. This engineered variant also showed a better resistance to pepsin under gastric conditions, allowing its clinical application in celiac disease therapy [154]. However, the existing wild-type or variant PEPs are only able to degrade residual gluten instead of the complex gluten. Therefore, continuous protein engineering for more potent gluten-detoxifying enzymes is required to realize celiac disease therapy for gluten-intolerant patients. 

## 6. Conclusions

This review explores the structure–function studies of PPCEs from different taxa and highlights many of their biological functions. PPCEs from family S9, S28, S33, and neprosin shared a common two-domain structure but with different catalytic and non-catalytic domains in the control of substrate binding and post-proline cleaving activity. Meanwhile, the zinc metallopeptidase families show diverse structures, such as that of M2 family with two metallopeptidase domains sandwiching the active channel with a bound zinc ion, which is different from the M12 family with N-terminal and C-terminal domains. PPCEs found in diverse organisms are useful in various ways given their capability to degrade proline-rich proteins. Moreover, enzyme characterization studies for both the native and recombinant PPCEs from various species suggest a promising potential for PPCEs to be applied according to their optimum conditions for various purposes. This review provides examples of extensive PPCE biotechnological applications in the beer-brewing industry, as potential therapeutic agents for individuals intolerant to gluten, as drug targets for various diseases, and contribute to proteomics analysis. Furthermore, inhibitors of PPCEs can be applied to type 2 diabetes treatment, memory learning enhancement, and prodrug cancer treatment. Many investigations have been conducted to find the ideal enzyme that can degrade gluten in small doses under gastric conditions, mostly focusing on microbial enzymes. More studies of PPCEs are needed in plants to understand their physiological roles for beneficial agricultural applications, such as nutrient mobilization and stress resistance. Hopefully, further protein engineering and structure-function studies with the aid of ab initio structural prediction will uncover more useful PPCEs for new biotechnological applications.

## Figures and Tables

**Figure 1 plants-11-01330-f001:**
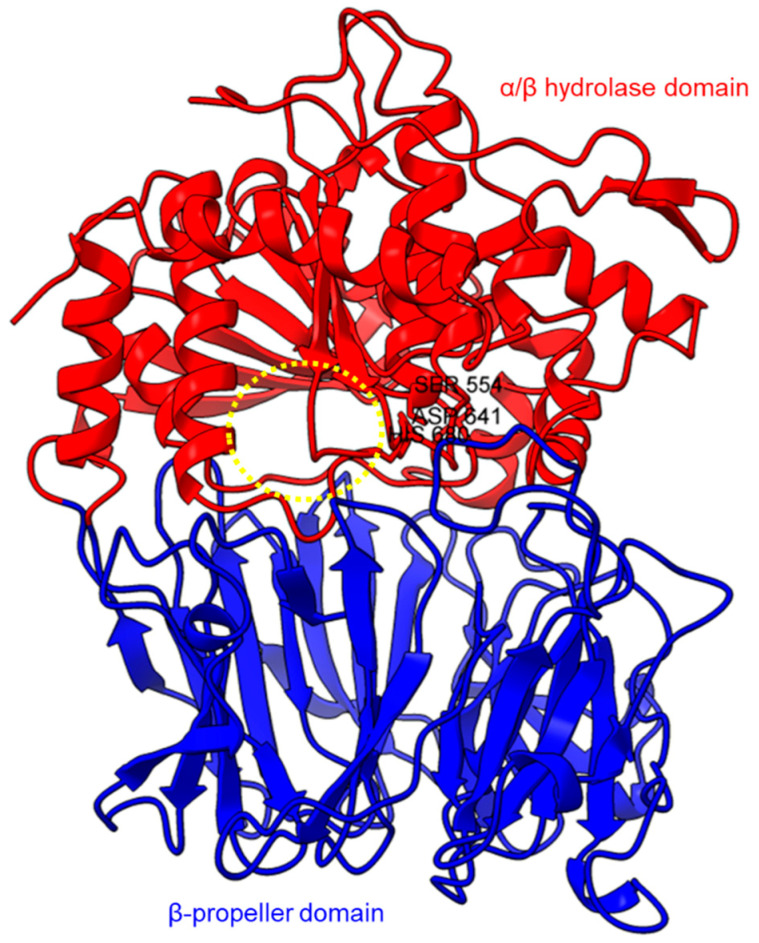
Crystal structure of a porcine POP (PDB ID: 1QFM). In red is the α/β hydrolase catalytic domain, whereas the β-propeller domain is highlighted in blue. Catalytic triad residues: Ser554, Asp641, and His680 are labeled. The yellow circle indicates a probable accessible path discovered in an electron microscopy study [75].

**Figure 2 plants-11-01330-f002:**
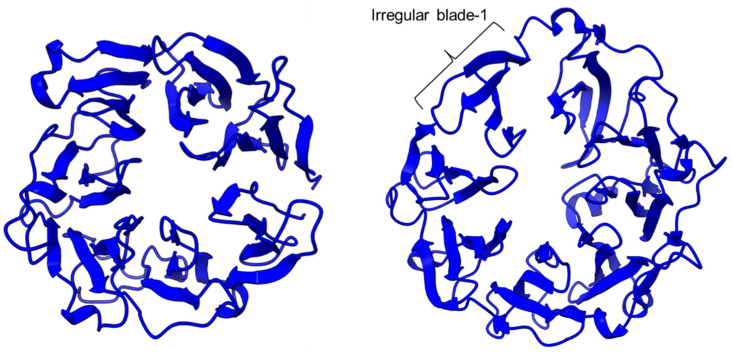
The β-propeller structure of a porcine prolyl oligopeptidase POP (PDB id: 1QFM, left) and a human dipeptidyl peptidase DPP-IV (PDB ID: 1J2E, right). The labeled irregular blade-1 of the eight-blade β-propeller structure could contribute to different substrate specificity between POP and DPP-IV.

**Figure 3 plants-11-01330-f003:**
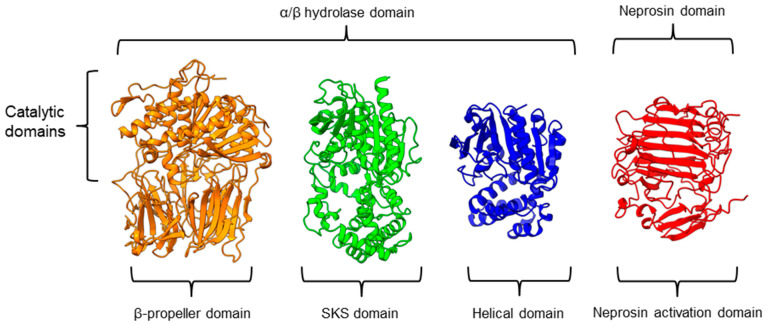
Representative protein structures from the MEROPS peptidase family S9 (porcine muscle prolyl oligopeptidase, PDB ID: 1QFM), S28 (*Aspergillus niger* prolyl endoprotease, PDB ID: 7WAB), S33 (*Serratia marcescens* prolyl aminopeptidase, PDB ID: 1QTR), and U74 (AlphaFold2 model of *N. × ventrata* neprosin, NvNpr), from left to right. The three-dimensional protein structures are oriented with their catalytic domains on top.

**Figure 4 plants-11-01330-f004:**
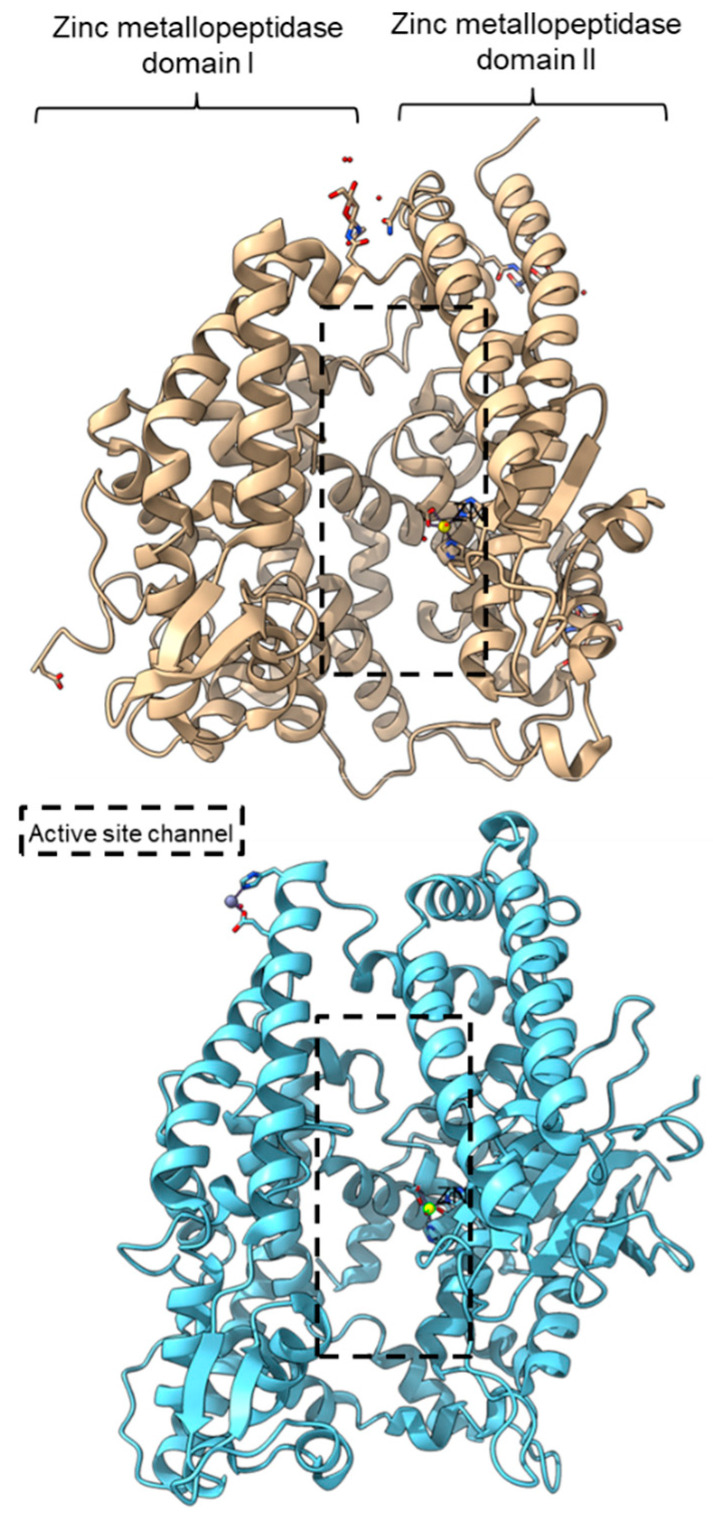
Crystal structures of the MEROPS metallopeptidase M02.006 angiotensin-converting enzyme-2, ACE2 (top, PDB ID: 1R42), and M03.002 neurolysin (bottom, PDB ID: 1I1I).

**Figure 5 plants-11-01330-f005:**
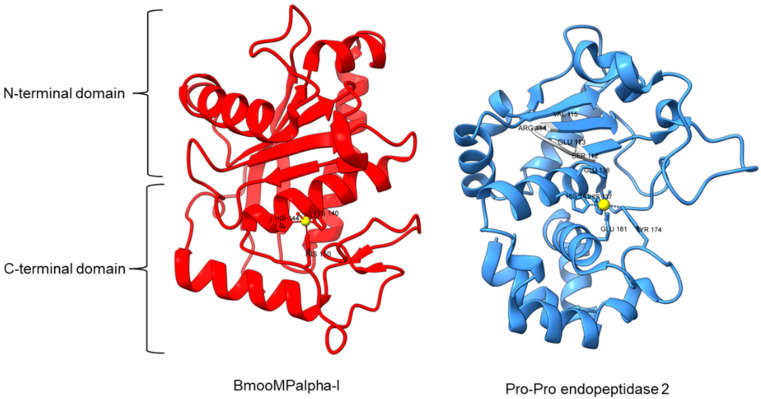
Crystal structures of MEROPS metallopeptidase M12.338 BmooMPalpha-I (left, PDB ID: 3GBO) and M34.003 Pro-Pro endopeptidase 2 (right, PDB ID: 6FPC) with substrate loop (S-loop) SERV in light gray. The zinc ions are shown as yellow spheres.

**Figure 6 plants-11-01330-f006:**
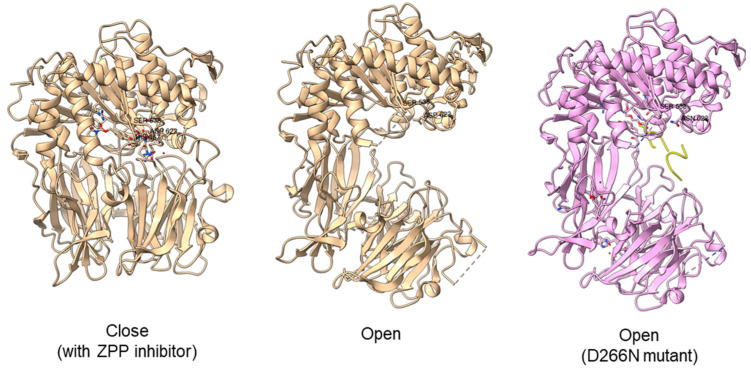
Crystal structures of *Aeromonas punctata* PEP (ApPEP). From left to right: close-state wild type ApPEP with ZPP inhibitor (PDB ID: 3IVM), open-state wild type ApPEP (PDB ID: 3IUL), mutant ApPEP with D266N mutation in an open state (PDB ID: 3IUR) induced by two H2H3 substrates (yellow). The catalytic triad (Ser538, Asp622, and His657) are labeled. However, the loops containing histidine catalytic residue (His657) are missing in the open-state crystal structures as they are unresolved in electron density maps and are inherently flexible.

## Data Availability

Not applicable.

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
