# Peer review of "Post-Proline Cleaving Enzymes (PPCEs): Classification, Structure, Molecular Properties, and Applications"

_plants, 2022, doi:10.3390/plants11101330_

Round 1
Reviewer 1 Report
Authors have written a review that summarises the post-proline cleaving enzymes. Authors have summarised 17 PPCEs in table 1. While I think this manuscript is very useful for experts in the field, I think it is a bit chaotic with no explanation on why we are touching on specific points. I would like more of an explanation on why are these particularly discussed.
The manuscript is divided in 4 sections:
I read the manuscript multiple times and I still struggle to understand what is in particular special about the PEP/POP enzymes, what do they interact with, how are they different between different species? I feel like certain statements are lacking context and are too general. If I have these questions after reading it multiple times that means that the structure and message of the review is not conveyed clearly.
Does MEROPS contain all the possible PEPs that are in the literature? Have there been a new ones added/discovered in recent years? How much efforts and potential is there in this field? Does Uniprot contain any additional ones? I would also like to see what are the interaction networks between these peptides and the mechanisms of action.
For the structural studies, (very large part of the review) I would like a clearer explanation on why we are discussing these 3D structures? What translational impact can this provide, and what potential this could bring? Why are these particular structures picked over other available and what’s common between them? I noticed that also authors touch on AlphaFold structures, can extra information can be revealed from this – in particular solvent accessibility?
Authors very briefly touch on different species with some specific examples, but I am unsure what is common and what’s different between these species for the particular type of proteases? Do they play the same function? What are the alignments between these species, and what are the active domains in each species, any conserved?
For the theuraputic agents – I would certainly want to know more about this? This is very general and does not provide me a lot of info. You could find an evidence of most of the proteases being involved in a human brain disease, this is very very general and does not give me anything new when I read it. This section needs to be much more specific. Do PEPs release any matrikines as they cleave their substrates? How do they work, what are the delivery mechanisms? How are these specific proteases used in applications and biotechnological efforts?
For the proteomics tool – Authors say PEP has a unique mechanism of action, but what is unique about it? Trypsin is generally used for its specificity in cleaving only of Arg and Lys residues, hence the post hoc search is easier to match the peptide spectra. If you want to generate a lot of peptides you could apply a very non-specific enzyme and the coverage will improve, but it will complicate the analysis. Peptides will certainly be unique as compared with trypsin because the cleavage motifs are different.
For the Protein Engineering I am surprised that authors only refer to a very specific example of modifying the certain residues. I would like to know of more of an opportunity in doing this, is there more efforts in this field? Has PEPs been used as a topical application, and has this been improved by any of the protein engineering efforts?
Conclusions are very general, what are the potentials for this type of peptidases? How can we utilise them in therapeutics, food applications agriculture?
Overall a good first effort, but the reviews should leave me with a clear understanding of the current applications instead of wondering what is it that I gained out of reading the review?
Thanks to authors tho. I certainly find this field very exciting!
Author Response
Author's Reply to the Review Report (Reviewer 1)
Authors have written a review that summarises the post-proline cleaving enzymes. Authors have summarised 17 PPCEs in table 1. While I think this manuscript is very useful for experts in the field, I think it is a bit chaotic with no explanation on why we are touching on specific points. I would like more of an explanation on why are these particularly discussed.
We thank Reviewer 1 for reviewing our manuscript that summarises the post-proline cleaving enzymes (PPCEs) from 17 MEROPS families of proteases. This review is intended to be broad in scope to cover PPCEs from diverse taxa, especially in their state-of-art structure-function studies to date not just for the experts but also for the general readers to have an overview of PPCEs. It is not our intention to go in-depth into specific families but rather to highlight the properties of different PPCEs from different families in general.
The manuscript is divided in 4 sections:
This review comprises 6 sections: Introduction, Molecular Structure and Biochemistry, Biological Functions of PPCEs, Biotechnological Applications, Protein Engineering, and Conclusion, in which the main 4 sections are organized and structured to cover the individual aspects of PPCEs for discussion based on the focus and scope of this review.
I read the manuscript multiple times and I still struggle to understand what is in particular special about the PEP/POP enzymes, what do they interact with, how are they different between different species? I feel like certain statements are lacking context and are too general. If I have these questions after reading it multiple times that means that the structure and message of the review is not conveyed clearly.
We apologize for the confusion. PPCEs represent diverse families of peptidases with post-proline cleaving (PPC) activity. Many of these PPCEs are from the PEP/POP serine family. PEP/POP enzymes are extensively studied in the biomedical field for their involvement in human pathology and neurological functioning in the brain. Different PEP/POP enzymes have different physiological substrates, which have been previously reviewed and cited in section 3.2 of the manuscript. In this current review, we have included the other families of peptidases with PPC activity apart from the S9 family of PEP/POP to showcase the different structures and functions of PPCEs. It is beyond the scope of this review to cover extensively the protein interactions and differences of specific families of peptidases between different species, in which the readers can refer to the previous reviews focusing on specific families of peptidases. We have made the aims and scope of this review clearer in the revised manuscript to cover only the biochemical properties, biological functions, and biotechnological applications of PPCEs from different families in the respective sections of the manuscript [Highlighted in the Abstract & Introduction].
Does MEROPS contain all the possible PEPs that are in the literature? Have there been a new ones added/discovered in recent years? How much efforts and potential is there in this field? Does Uniprot contain any additional ones? I would also like to see what are the interaction networks between these peptides and the mechanisms of action.
To date, the MEROPS database is the most comprehensive database for all peptidases, which is manually curated and updated monthly [1]. As mentioned above and in the caption of Table 1, the list of PPCEs is non-exhaustive but this review included studies of PEPs found in MEROPS as well as those in the literature for discussion. To clarify, PPCEs encompass PEPs and other peptidases with PPC activity. PEPs or POPs are generally referred to as the serine peptidase family. There have been new peptidases discovered in recent years with PPC activity, which have motivated the inception of this review. The curation of peptidases are continuous as exemplified by the great efforts of MEROPS curator. Our analysis of PPCEs was based on the latest version of MEROPS database search (https://www.ebi.ac.uk/merops/cgi-bin/specsearch.pl) of 108,005 cleavage site records with query XXX-XXX-XXX-PRO|XXX-XXX-XXX-XXX to identify the specificity of post-proline cleavage of different peptidase families (Table 1).
UniProt will definitely contain other uncurated PPCEs from transcriptomics or proteomics studies that show homology to reported PPCEs. However, it is not our aim to provide an exhaustive list of all possible PPCEs in the database/literature. Rather, this review provides the first attempt to discuss the structure-function analysis of PPCEs from diverse taxa and showcase the presence of PPCEs from different peptidase families besides the serine peptidases. The discovery of neprosin from Nepenthes spesies is a good example of a new PPCE with great potential for celiac disease therapy, which doesn’t fit into the extensively studied serine family of PEP/POP. The prolyl endopeptidase of Aspergillus niger from MEROPS family S28 is another PPCE that has promising potential in the therapeutic and food processing industries.
The interactions of PEP with other proteins have been extensively reviewed for animals/humans, such as this recent PEP review [2], which summarized proteins that interact with PEP including α-tubulin, α-synuclein, GAP-43, and α-2-macroglobulin. No such studies for PPCEs of other taxa. We have included brief descriptions of the protein-protein interactions (PPIs) of PEPs in the introduction, pointing to their physiological roles in the biological system. We have also included the reported PPIs of a PEP with its protein partners with a brief discussion in section 4.2. We highlighted that PEP may act as a regulator, while the mechanism and regulation of PPIs remain a knowledge gap to be addressed. However, extensive descriptions on the interaction networks between these peptides and the mechanisms of action will be beyond the scope of this review as stated above.
For the structural studies, (very large part of the review) I would like a clearer explanation on why we are discussing these 3D structures? What translational impact can this provide, and what potential this could bring? Why are these particular structures picked over other available and what’s common between them? I noticed that also authors touch on AlphaFold structures, can extra information can be revealed from this – in particular solvent accessibility?
This review compiles and compares some of the well-studied PPCEs from different families of MEROPS peptidase, which to our knowledge is the first-ever attempt. The purpose of the section on structural studies is to show the similarities and differences of structures from different families of PPCEs, which we have clarified in the revised manuscript. We highlight the common features of PPCEs from families S9, S28, S33, and U74 which is a two-domain structure, and depict their respective catalytic domains in Figure 3, as well as summarize the function of non-catalytic domains (Lines 188-195). The discussion of PPCE structures (section 2.1) serves as a summary of PPCEs structural information for future reference.
The particular protein structures are chosen because they are the representative member of PPCEs of each family with protein crystal structures available on the Protein Database (PDB). We touch on AlphaFold2 because the protein structure of neprosin is obtained through this protein modeling method. There is no crystal structure of neprosin available to date and AlphaFold2 is a state-of-art ab initio protein modeling technology released in 2021 that is capable of predicting experimental-level protein structure [3]. The AlphaFold2 model of neprosin is included because i) the model quality is of experimental quality, ii) neprosin is an interesting plant PPCE with very promising potential in celiac disease treatment [4] and proteomics study, especially in histone mapping [5,6]. However, further discussion on the applications of AlphaFold2 such as the solvent accessibility is beyond the scope of this review.
Authors very briefly touch on different species with some specific examples, but I am unsure what is common and what’s different between these species for the particular type of proteases? Do they play the same function? What are the alignments between these species, and what are the active domains in each species, any conserved?
We mentioned that PPCEs even if they have inherently different protein structures from different clans such as serine, glutamic, and metallopeptidase can have post-proline cleaving activity. Not all functions of PPCEs are known, thus we discussed the reported functions of PPCEs in section 3 (Biological Functions of PPCEs). The PPCEs are categorized into different families in the MEROPS database, and each family has its functional domain as we discussed in section 2.1 (Structural Studies). The alignments of PPCEs (amino acid sequences or structural alignment) from different families will not yield further insights for this review since PPCEs have been categorized into different families in the MEROPS database based on sequence homology and shared common ancestry. For details on differences between species for a particular type of protease, the readers can refer to the previous reviews on specific families of peptidases. Hope the reviewer understands that this review aims to provide a bird’s-eye view of various PPCE families rather than go in-depth into all the specific families.
For the theuraputic agents – I would certainly want to know more about this? This is very general and does not provide me a lot of info. You could find an evidence of most of the proteases being involved in a human brain disease, this is very very general and does not give me anything new when I read it. This section needs to be much more specific. Do PEPs release any matrikines as they cleave their substrates? How do they work, what are the delivery mechanisms? How are these specific proteases used in applications and biotechnological efforts?
We did not discuss in-depth how PEP is involved in human brain disease as the exact mechanism of PEP causing the disease is not yet clear. Besides, there are well-written reviews of PEP, human brain diseases, and PEP inhibitors published, which we have referenced in this review [2,7,8]. In the revised manuscript, we have included a statement that there is no consensus on PEP causing human cognitive diseases with supporting references in Section 4.2.
The scenario where PEPs release matrikines is the proinflammatory POPs that are involved in the pathology of lung disease. In that case, the proinflammatory POPs together with matrix metalloproteinases (MMPs) can generate matrikine proline-glycine-proline (PGP) by breaking down proline-rich collagen of the extracellular matrix (ECM). The end-product of POP cleavage, prolyl-glycyl-proline (PGP) is then acetylated into AcPGP, which acts as a chemoattractant for neutrophil recruitment and continuously drives the cycle of neutrophilic inflammation. This statement has been added to Section 3.2.
Further discussion on how the therapeutics work, delivery mechanisms, and how are these specific proteases used in applications and biotechnological efforts warrant another standalone review.
For the proteomics tool – Authors say PEP has a unique mechanism of action, but what is unique about it? Trypsin is generally used for its specificity in cleaving only of Arg and Lys residues, hence the post hoc search is easier to match the peptide spectra. If you want to generate a lot of peptides you could apply a very non-specific enzyme and the coverage will improve, but it will complicate the analysis. Peptides will certainly be unique as compared with trypsin because the cleavage motifs are different.
We have thoroughly revised Section 4.3 to make our points clearer in the revised manuscript. PEPs have a unique mechanism of action by cleaving substrate in a different pattern as compared to trypsin in proteomics. PEPs cleave substrate by attacking the Pro-X bond resulting in different fragments providing more sequence coverage hence uncovering more information on the sequence of the particular protein. For example, neprosin generates peptides 1–38 of histone H3 and peptides 1–32 of histone H4 in a single digest of Hela cells, permitting the analysis of co-occurring post-translational modifications in these important N-terminal tails [6]. Neprosin is not to be used to replace the standard protease trypsin in proteomics but applied complementarily for proteomic analysis. Histone mapping studies [5,6] have demonstrated the useful application of neprosin in mass spectrometry analysis with more specific PPC activity than AN-PEP [6].
For the Protein Engineering I am surprised that authors only refer to a very specific example of modifying the certain residues. I would like to know of more of an opportunity in doing this, is there more efforts in this field? Has PEPs been used as a topical application, and has this been improved by any of the protein engineering efforts?
We have included further protein engineering efforts on the engineering of human prolyl endopeptidase for Antibody-Directed Enzyme Prodrug Therapy and the co-administration of cysteine endoprotease EP-B2 with improved mutagenesis variants of PEP from Sphingomonas capsulate for improving gluten-detoxification activity in section 5 (Protein Engineering).
Again, the in-depth review of the opportunity and bioengineering efforts in the biotechnological applications of PEPs, such as their topical application and improvement via protein engineering efforts (if any) should warrant another standalone review that is beyond the scope and goal of this review in providing an overview of the investigations related to PPCEs for further studies.
Conclusions are very general, what are the potentials for this type of peptidases? How can we utilise them in therapeutics, food applications agriculture?
The conclusion summarizes take-home meassages from the respective sections of this review, which addressed the many potential applications in therapeutics, food applications, and agriculture as explained in the revised conclusion:
This review explores the structure-function studies of PPCEs from different taxa and highlights many of their biological functions. PPCEs from family S9, S28, S33, and neprosin shared a common two-domain structure but with different catalytic and non-catalytic domains in the control of substrate binding and post-proline cleaving activity. Meanwhile, the zinc metallopeptidase families show diverse structures, such as that of M2 family with two metallopeptidase domains sandwiching the active channel with a bound zinc ion, which is different from the M12 family with N-terminal and C-terminal domains. PPCEs found in diverse organisms are useful in various ways given their capability to degrade proline-rich proteins. Enzyme characterization studies for both the native and recombinant PPCEs from various species suggest a promising potential for PPCEs to be applied according to their optimum conditions for various purposes. This review provides examples of extensive PPCE biotechnological applications in the beer-brewing industry, as potential therapeutic agents for individuals intolerant to gluten, as drug targets for various diseases, as well as applications in proteomics analysis (Section 4). Moreover, many efforts have been done in using inhibitors of PPCEs for type 2 diabetes treatment, memory learning enhancement, and prodrug cancer treatment. Many researchers are working hard in finding the ideal enzyme that can degrade gluten in small doses under gastric stomach simulations. More PPCE studies are needed in plants to understand their physiological roles for beneficial agricultural applications, such as nutrient mobilization and stress resistance. Hopefully, further protein engineering and structure-function studies with the aid of ab initio structural prediction will uncover more useful PPCEs for new biotechnological applications.
Overall a good first effort, but the reviews should leave me with a clear understanding of the current applications instead of wondering what is it that I gained out of reading the review?
We hope that the reviewer will find the revised manuscript clearer in the current applications of PPCEs and understand the purpose of this review.
Thanks to authors tho. I certainly find this field very exciting!
We are glad that the reviewer was inspired by this review, which achieves the goal of this review in providing an overview of PPCE studies to date to encourage further investigations on this exciting field of PPCEs. We apologize if we couldn’t include all the content required by Reviewer 1 to keep the review to the scope that we intended.
References
- Rawlings, N.D.; Barrett, A.J.; Thomas, P.D.; Huang, X.; Bateman, A.; Finn, R.D. The MEROPS database of proteolytic enzymes, their substrates and inhibitors in 2017 and a comparison with peptidases in the PANTHER database. Nucleic Acids Research 2018, 46, 624-632, doi:10.1093/nar/gkx1134.
- Männistö, P.T.; García-Horsman, J.A. Mechanism of Action of Prolyl Oligopeptidase (PREP) in Degenerative Brain Diseases: Has Peptidase Activity Only a Modulatory Role on the Interactions of PREP with Proteins? Frontiers in Aging Neuroscience 2017, 9, doi:10.3389/fnagi.2017.00027.
- Jumper, J.; Evans, R.; Pritzel, A.; Green, T.; Figurnov, M.; Ronneberger, O.; Tunyasuvunakool, K.; Bates, R.; Žídek, A.; Potapenko, A.; et al. Highly accurate protein structure prediction with AlphaFold. Nature 2021, 596, 583-589, doi:10.1038/s41586-021-03819-2.
- Rey, M.; Yang, M.; Lee, L.; Zhang, Y.; Sheff, J.G.; Sensen, C.W.; Mrazek, H.; Halada, P.; Man, P.; McCarville, J.L.; et al. Addressing proteolytic efficiency in enzymatic degradation therapy for celiac disease. Sci. Rep. 2016, 6, doi:10.1038/srep30980.
- Schräder, C.U.; Ziemianowicz, D.S.; Merx, K.; Schriemer, D.C. Simultaneous Proteoform Analysis of Histones H3 and H4 with a Simplified Middle-Down Proteomics Method. Anal. Chem. 2018, 90, 3083-3090, doi:10.1021/acs.analchem.7b03948.
- Schräder, C.U.; Lee, L.; Rey, M.; Sarpe, V.; Man, P.; Sharma, S.; Zabrouskov, V.; Larsen, B.; Schriemer, D.C. Neprosin, a selective prolyl endoprotease for bottom-up proteomics and histone mapping. Molecular & cellular proteomics : MCP 2017, 16, 1162-1171, doi:10.1074/mcp.M116.066803.
- Babkova, K.; Korabecny, J.; Soukup, O.; Nepovimova, E.; Jun, D.; Kuca, K. Prolyl oligopeptidase and its role in the organism: attention to the most promising and clinically relevant inhibitors. Future medicinal chemistry 2017, 9, 1015-1038, doi:10.4155/fmc-2017-0030.
- Gass, J.; Khosla, C. Prolyl endopeptidases. Cellular and molecular life sciences : CMLS 2007, 64, 345-355, doi:10.1007/s00018-006-6317-y.
Reviewer 2 Report
The manuscript proposed by Baharin and co-workers (plants-1695859) entitled “Post-proline cleaving enzymes (PPCEs): Classification, structure, molecular properties, and applications” is the first comprehensive review to cover the biotechnological applications of PPCEs and discuss the unique prolyl cleaving activity of different enzymes based on the recent structure-function studies from diverse taxa. In my opinion, the presented manuscript needs some minor changes and corrections.
My comments are presented below.
Major concerns:
- Abstract – clearly present the importance of post-proline cleavage.
- Introduction – short paragraph presenting how often the post-proline proteolysis can be found is needed. Additionally, once again present the importance of post-proline cleavage.
- Short paragraph presenting the substrate specificity of post-proline cleavages enzymes and discussion is needed
- In the analysis of substrate specificity of enzymes, both proteinogenic and unnatural amino acids are used. Give some information about the application of unnatural amino acids in the analysis of substrate specificity of post-proline cleavage enzymes
- give information about the methods used in the analysis of substrate specificity of PPCEs
- 4.3 Proteomics tool, page 21, lines 203-2019 – discuss or suggest how commonly available tags for quantitative or qualitative proteomics may work in the case of post-proline cleavage. What about the mass spectra interpretation, what about the fragment ion formation? Will proline effect play a crucial role in it?
- present some possible future aspects of the application of described PPCEs
Check and correct the reference style according to the journal guide (i.e. ref 8, 12, 20, 63)
Paper presented in Preprints.
Make changes in the text.
Check and correct English
Author Response
We thank Reviewer 2 for reading our manuscript with constructive comments!
- Abstract – clearly present the importance of post-proline cleavage.
We have included the following statement in the abstract: “PPCEs display preferences towards the Pro-X bonds for hydrolysis. This level of selectivity is substantial and has benefited the brewing industry, therapeutics for celiac disease by targeting proline-rich substrates, drug targets for human diseases, and proteomics analysis.”
- Introduction – short paragraph presenting how often the post-proline proteolysis can be found is needed. Additionally, once again present the importance of post-proline cleavage.
We have included a short paragraph in the introduction to highlight the prevalence and importance of post-proline proteolysis (Lines 70-77). The specificity of post-proline cleavage is discussed further below.
- Short paragraph presenting the substrate specificity of post-proline cleavages enzymes and discussion is needed
We have included the descriptions on the the substrate specificity of post-proline cleavages enzymes in the introduction (Lines 55-60) as well as discussed in Section 2.3 Biochemical Studies of PPCEs (Lines 317-337).
- In the analysis of substrate specificity of enzymes, both proteinogenic and unnatural amino acids are used. Give some information about the application of unnatural amino acids in the analysis of substrate specificity of post-proline cleavage enzymes
Proteinogenic and unnatural amino acids have been used by various researchers to analyze the substrate specificity of post-proline cleavage enzymes. This part has been elaborated in the revised Section 2.3 (Lines 304-337).
- give information about the methods used in the analysis of substrate specificity of PPCEs
The methods used are listed in Table 3 (column “substrate” and “method to detect digested substrate”) and Table 4 (column “substrate” and “assay detection”). Specific studies have been done experimentally to identify the substrate specificity of these proteases including the usage of mass spectrometry and fluorogenic substrates. Chromogenic substrates designed with a chromogenic component such as pNa and βNA after proline residue are often used to detect and determine post-proline cleaving activity of PPCEs. This part has been elaborated in the revised Section 2.3 (Lines 304-337).
- 4.3 Proteomics tool, page 21, lines 203-2019 – discuss or suggest how commonly available tags for quantitative or qualitative proteomics may work in the case of post-proline cleavage. What about the mass spectra interpretation, what about the fragment ion formation? Will proline effect play a crucial role in it?
Not many works in the past years describe the quantitative or qualitative proteomics approach using PPCEs outside of AN-PEP and neprosin. There is no information on how currently available tags would work with PPCEs. Current work mainly focuses on AN-PEP as a complementary protease for mass spectrometry and neprosin in histone mapping. We have added the latest research paper using neprosin as a new simplified middle-down approach for histone mapping. Section 4.3 (Lines 594-636) has been rewritten for clarity. There is no study performed on the effect of proline in mass spectra interpretation and fragment ion formation. This topic could be an interesting topic for further investigation.
- present some possible future aspects of the application of described PPCEs
We have included some of the future aspects of PPCEs in Section 6 (conclusion) Lines 688-694.
Check and correct the reference style according to the journal guide (i.e. ref 8, 12, 20, 63)
The reference style have been corrected according to the journal guideline.
Round 2
Reviewer 1 Report
Happy with my comments being adressed, thank you authors.